# Entropy based C4.5-SHO algorithm with information gain optimization in data mining

G Sekhar Reddy[1] and Suneetha Chittineni[2]

[1] Department of Computer Science and Engineering, Acharya Nagarjuna University, Guntur, Andhra Pradesh, India
[2] Department of Computer Applications, RVR&JC college of Engineering, Guntur, Andhra Pradesh, India



## ABSTRACT

Information efficiency is gaining more importance in the development as well as application sectors of information technology. Data mining is a computer-assisted process of massive data investigation that extracts meaningful information from the datasets. The mined information is used in decision-making to understand the behavior of each attribute. Therefore, a new classification algorithm is introduced in this paper to improve information management. The classical C4.5 decision tree approach is combined with the Selfish Herd Optimization (SHO) algorithm to tune the gain of given datasets. The optimal weights for the information gain will be updated based on SHO. Further, the dataset is partitioned into two classes based on quadratic entropy calculation and information gain. Decision tree gain optimization is the main aim of our proposed C4.5-SHO method. The robustness of the proposed method is evaluated on various datasets and compared with classifiers, such as ID3 and CART. The accuracy and area under the receiver operating characteristic curve parameters are estimated and compared with existing algorithms like ant colony optimization, particle swarm optimization and cuckoo search.

## INTRODUCTION

Information management is comprised of mining the information, managing data warehouses, visualizing the data, knowledge extraction from data and so on (*Chen et al., 2018*). Consequently, different information management techniques are now being applied to manage the data to be analyzed. Hence, it is necessary to create repositories and consolidate data as well as warehouses. However, most of the data may be unstable; so it is essential to decide the data to be stored and discarded (*Amin, Chiam & Varathan, 2019*). In addition, individual storage is required to manage real-time data to conduct research and predict trends. Data mining techniques are becoming more popular, recently getting attention towards rule mining methods, such as link analysis, clustering and association rule mining (*Elmaizi et al., 2019*). Data mining discovers the substantial information, reasons and possible rules from huge datasets. It stands as an important

Corresponding author
G Sekhar Reddy,
golamari.sekhar@gmail.com

source for information system based decision-making processes, such as classification, machine learning and so on (*Sun et al., 2019a*). Data mining is generally a specific term to define certain computational analysis and results that comply with three main properties like comprehension, accuracy and user requirements. Data mining techniques are very useful while dealing with large datasets having vast amount of data. The data mining research community has been active for many years in analyzing various techniques and different applications of data mining (*Jadhav, He & Jenkins, 2018*).

A system that is combined with both data analysis and classification is suggested to create mining rules for several applications. For extracting the relevant information from systems, functional knowledge or rules automatically activates the mining process to provide rapid, real-time and significant operational basis. The classification approaches broadly used in data mining applications is efficient in processing large datasets (*Gu et al., 2018*). It maps an input data object into one of the pre-defined classes. Therefore, a classification model must be established for the given classification problem (*Junior & Do Carmo Nicoletti, 2019*). To perform the classification task, the dataset is converted into several target classes. The classification approach assigns a target type for each event of the data and allots the class label to a set of unclassified cases. This process is called supervised learning because all the training data are assigned as class tags. Therefore, classification is used to refer the data items as various pre-defined classes (*Xie et al., 2020*). The classifier is categorized into two approaches namely logical reasoning and statistical analysis. To create a well-trained classifier, training data are used to signify the key features of the classification problem under analysis (*Meng et al., 2020*). Once the classifier is trained, then the test dataset is evaluated by the classifier. The overall performance of any classifier algorithm is comparatively estimated through the sensitivities of minority target classes. However, the minority target class predictions are usually found below optimal because of the initial algorithm designs that consider identical class distribution in both model and usage (*Ebenuwa et al., 2019*).

The most popular and simple classification technique is the decision tree. Decision trees are popular learning tool utilized in functional research, especially in results analysis to achieve a goal. As a general logical model, a decision tree repeats the given training data to create hierarchical classification (*Es-sabery & Hair, 2020*). It is a simplest form of classifier that can be stored densely and effectively in order to categorize the new data. It takes inputs in the form of training data set, attribute list and attribute selection method. A tree node is created by the algorithm in which attribute selection is applied to compute optimal splitting criteria. Then the final node generated is named based on the selected attributes (*Damanik et al., 2019*). The training tuples subset is formed to split the attributes. Hence, parameters (like purity, number of samples, etc.) are still needed for a decision tree. Moreover, it is capable of handling multidimensional data that offers good classification performance for common datasets (*Ngoc et al., 2019*). The decision tree is also known as decision support tool which utilizes the model of tree-like or graph and the consequences are resource costs, utility and event outcomes (*Lee, 2019*). In practical terms, the methods utilized to create decision trees normally produce trees with a low node factor and modest tests at each node. Also, the classifier contains different algorithms,

such as C4.5, ID3 and CART. The C4.5 algorithm is the successor of ID3 which uses gain ratio by splitting criterion for splitting the dataset. The information gain measure used as a split criterion in ID3 is biased to experiments with multiple outcomes as it desires to select attributes with higher number of values (*Jiménez et al., 2019*). To overcome this, the C4.5 algorithm undergoes information gain normalization using split information value which in turn avoids over fitting errors as well.

In C4.5, two criterions are used to rank the possible tests. The first criterion of information gain is to minimize the entropy of subsets and the second criterion of gain ratio is to divide the information gain with the help of test outcome information. As a result, the attributes might be nominal or numeric to determine the format of test outcomes (*Kuncheva et al., 2019*). On the other hand, the C4.5 algorithm is also a prominent algorithm for data mining employed for various purposes. The generic decision tree method is created default for balanced datasets; so it can deal with imbalanced data too (*Lakshmanaprabu et al., 2019*). The traditional methods for balanced dataset when used for imbalanced datasets cause low sensitivity and bias to the majority classes (*Lakshmanaprabu et al., 2019*). Some of the imbalance class problems include image annotations, anomaly detection, detecting oil spills from satellite images, spam filtering, software defect prediction, etc (*Li et al., 2018*). The imbalanced dataset problem is seen as a classification problem where class priorities are very unequal and unbalanced. In this imbalance issue, a majority class has larger pre-probability than the minority class (*Liu, Zhou & Liu, 2019*). When this problem occurs, the classification accuracy of the minority class might be disappointing (*Tang & Chen, 2019*). Thus, the aim of the proposed work is to attain high accuracy in addition to high efficiency.

In data classification, accuracy is the main challenge of all applications. Information loss in dataset is problematic during attribute evaluation and so, the probability of attribute density is estimated. For this, the information theory called entropy based gain concept is utilized to enhance the classification task. Furthermore, common uncertainties of numerical data are used to measure the decision systems. A population based algorithm is utilized to optimize the gain attributes and to enhance the classification in complex datasets. The Selfish Herd Optimization (SHO) enhances the feature learning accuracy by effectively removing redundant features thereby providing good global search capability. The main contribution of the proposed work is summarized as follows.

- To solve the data classification problem using entropy based C4.5 decision tree approach and gain estimation.
- SHO algorithm is utilized to optimize the information gain attributes of decision tree.
- The data are classified with high accuracy and area under the receiver operating characteristic curve (AUROC) of datasets is compared with existing techniques.

The organization of this paper is described as follows: introduction about the research paper is presented in "Introduction", survey on existing methods and challenges are depicted in "Related Works". The preliminaries are explained in "Preliminaries". The working of proposed method is detailed in "Proposed Method: Selfish Herd

Optimization". Efficiency of optimization algorithm is evaluated in "Result and Discussion" and the conclusion of the proposed method is presented in "Conclusion".

## RELATED WORKS

Multiple learning process and multi-label datasets are widely used in different fields nowadays. *Yahya (2019)* evaluated the efficacy of Particle Swarm Classification (PSC) in data mining. PSC was utilized to design the classification model which classifies the queries into Bloom's taxonomy six cognitive-levels. Rocchio algorithm (RA) was used to mitigate the dimensionality of adverse effects in PSC. Finally, RA-based PSC was investigated with various feature selection (FS) methods for a scheme of queries. But it is identified that the multi-label classification dealt with some problems where the classifier chain label order has a strong effect on the performance of classification. Nevertheless, it is too hard to find the proper order of chain sequences. Hence, (*Sun et al., 2019b*) had proposed an ordering method based on the conditional entropy of labels where a single order was generated by this method. Reduced attributes can improve the accuracy of classification performances. The missed attribute values were typically not used in entropy or gain calculation. Information gain based algorithms tend to authenticate the attribute sets. Various measures were certainly affected from redundancy and non-monotonicity during attribute reduction. Therefore, a forward heuristic attribute reduction algorithm was proposed to solve the uncertainties in attribute selection. It simultaneously selects information attributes though unnecessary attributes were reduced in practice. *Gao et al. (2019)* proposed granular maximum decision entropy based on the measurement of monotonic uncertainty. Extreme decision entropy was developed in which the uncertainties of entropy are integrated with granulation knowledge. This investigation was validated with various UCI datasets and found to be computationally inexpensive.

The choice of dataset selection allows the extraction of highly representative information from high-level data; so computational efforts were reduced among other tasks. A hybrid optimization based FS was proposed by *Ibrahim et al. (2019)*. The suggested technique is combined with slap swarm algorithm (SSA) and particle swarm optimization methods to enhance the efficacy of global and local search steps. Therefore, the hybrid algorithm was examined on mixed datasets. It requires less time while the nodes quantity is reduced making it more desirable for large datasets. The SSA–PSO was employed to acquire best features from various UCI datasets. Also, redundant features were detached from the original datasets resulting in better accuracy. However, the accuracy is affected in complex datasets. To improve the classification performance of complex data, (*Sun et al., 2019c*) introduced an attribute reduction method utilizing neighborhood entropy measures. The systems should have the ability to handle continuous data while maintaining its information on attribute classification. The concept of neighborhood entropy was explored to deal with uncertainty and noise of neighborhood systems. It fully reflects the decision-making ability by combining the degree of reliability with the coverage degree of neighborhood systems.

A clustering method based on functional value sequences has been proposed to accurately identify the functional equivalent programs with index variations. Because existing clustering programs were limited to structured metric vectors as in *Wang et al. (2020)*. This strategy is implemented for automated program repair to identify the sample programs from a large set of template programs. The average accuracy and average entropy were 0.95576 and 0.15497, respectively. However, the problem turned out to uncertain as the number of predictions is higher than the number of previous results. This issue was overcome by an alternative solution of priori weights and maximum entropy principle to attain the posteriori weights. *Arellano, Bory-Reyes & Hernandez-Simon (2018)* utilized a machine learning approach with single aggregated prediction from a set of individual predictions. A new factor presents a problem departing from the well-known maximal entropy hypothetical method and taking the distance among original and estimated integrated predictions. The suggested method was applied to estimate and measure predictive capability using prediction datasets.

It is difficult to perform FS for multi-label dimension curse in numerous learning processes. Hence, *Paniri, Dowlatshahi & Nezamabadi-pour (2020)* proposed a multi-label relevance–redundancy FS scheme based on Ant Colony Optimization (ACO) called ML-ACO. ML-ACO seeks to find the best features with lowest redundancy and many repetitions with class labels. To speed up the convergence, the cosine similarities between features as well as class labels are used as starting pheromone for each ant, and can be classified as a filter-based method. Various parametric entropies of decision tree algorithms are investigated by *Bretó et al. (2019)*. Partial empirical evidences were provided to support the notion that parameter adjustment of different entropy activities influences the classification. Receiver operating characteristic (ROC) and AUROC curve analysis provides an accurate criterion for evaluating decision trees based on parametric entropy. Various entropies, such as Shannon entropy, Renyi entropy, Tsallis entropy, Abe entropy and Landsberg–Vedral entropy were discussed.

A new information classification algorithm has been introduced to improve the information management of restricted properties in *Wang et al. (2019)*. Information management efficiency has gained more importance for the development of information technology through its expanded use. Reduce leaf based on optimization ratio algorithm was utilized to optimize the decision tree ratios. ID3 algorithm is a classical method of data mining that selects attributes with maximum information gain from the dataset at split node. However, decision tree algorithms have some drawbacks; it is not always optimal and it is biased in favor of properties that have higher values. In data classification, accuracy is the main challenge of all datasets. The resulting information loss is problematic for attribute evaluation while estimating the probability density of attributes. Due to the absence of classification information, it is challenging to perform potential classification. Consequently, an improved algorithm is utilized to solve the data classification issues.

## PRELIMINARIES

Entropy based measurements understands the decision system knowledge, its properties and some relations about the measurements. An optimization model is explored to

enhance the performance of complex dataset classification. During prediction, the information gain optimal weights will be updated with the help of SHO algorithm. The nominal attributes of the dataset were designed by the ID3 algorithm. The attributes with missing values are not permitted. C4.5 algorithm, an extension of ID3 can handle datasets with unknown-values, numeric and nominal attributes (*Agrawal & Gupta, 2013*). C4.5 is one of the best learning based decision tree algorithm in data mining because of its distinctive features like classifying continuous attributes, deriving rules, handling missing values and so on (*Wu et al., 2008*). In decision tree based classification, the training set is assumed as $M$ and the number of training samples is mentioned as $|M|$. Here, the samples are divided into $N$ for various kinds of $K_1, K_2, ....K_n$ where the class sizes are labeled into $|K_1|, |K_2|, ...|K_n|$. A set of training sample is denoted as $M$, and the sample probability formula of class $K_i$ is given in Eq. (1).

$$p(M_i) = \frac{|K_i|}{|M|} \tag{1}$$

## Quadratic entropy

Entropy is used to measure the uncertainty of a class using the probability of particular event or attribute. The gain is inversely proportional to entropy. The information gain is normally dependent on the facts of how much information was offered before knowing the attribute value and after knowing the attribute value. Different types of entropies are utilized in data classification. For a better performance, quadratic entropy is used in our work (*Adewole & Udeh, 2018*). This entropy considers a random variable $X$ as finite discrete with complete probability collection as mentioned in Eq. (2).

$$p_i \geq 0 (i = 1, 2, ...n), \sum_{i=1}^{k} p_i = 1 \tag{2}$$

Here, the probability of event is denoted as $p_i$. The quadratic entropy of information is calculated by Eq. (3).

$$\text{Entropy } M(x) = \sum_{i=1}^{n} p_i(1 - p_i) \tag{3}$$

Here, $(M)$ specifies the information entropy of $M$ (training sample set). For this particular attribute, the entropy of information is determined by Eq. (4).

$$\text{entropy}(M, H) = \sum_{g \in G} \left( \frac{|M_g|}{|M|} \right) * \text{entropy}(M_g) \tag{4}$$

The entropy of attribute $H$ is represented by Entropy$(M, H)$, where $H$ signifies attribute value. $G$ Denotes all sets of values of $g$ and $M_g$ denotes the subset of $M$ which is the value of $H$. $|M_g|$ denotes the number of elements in $M_g$, and number of elements of $|M|$ in $M$.

## Information gain

The information gain is determined by Eq. (5).

$$\text{gain}(M, H) = \text{entropy}(M) - \text{entropy}(M, H) \tag{5}$$

In a dataset $M$, Gain $(M, H)$ denotes the information gain of attribute $H$. Entropy $(M)$ signifies the sample set of information entropy and Entropy $(M, H)$ denotes the information entropy of attribute $H$. In Eq. (5), information gain is employed to find additional information that provides high information gain on classification. C4.5 algorithm chooses the attribute that has high gain in the dataset and use as the split node attribute. Based on the attribute value, the data subgroup is subdivided and the information gain of each subgroup is recalculated. The decision tree trained process is enormous and deep compared to neural networks, such as KNN, ANN and etc. as it does not take into account the number of leaf nodes. Moreover, the gain ratio is different from information gain. Gain ratio measures the information related to classification obtained on the basis of same partition. C4.5 uses the information gain and allows measuring a gain ratio. Gain ratio is described in Eq. (6).

$$\text{gain\_ratio}(M, H) = \frac{\text{gain}(M, H)}{\text{split\_inf } o(M, H)} \tag{6}$$

where,

$$\text{split\_inf } o(M, H) = \sum_{g=1}^{n} -\frac{M_g}{M} \log_2 \frac{M_g}{M} \tag{7}$$

The attribute with a maximum gain rate is selected for splitting the attributes. When the split information tactics is 0, the ratio becomes volatile. A constraint is added to avoid this, whereby the information gain of the test selected must be large at least as great as the average gain over all tests examined.

## C4.5 decision tree

*Quinlan (2014)* developed the C4.5 algorithm to generate a decision tree. Many scholars have made various improvements in the tree algorithm. However, the problem is that tree algorithms require multiple scanning and deployment of data collection during the building process of decision trees. For example, large datasets provided into the ID3 algorithm improves the performance but not effective whereas small datasets are more effective in several fields like assessing prospective growth opportunities, demographic data, etc. This is because the processing speed is slow and the larger dataset is too large to fit into the memory. Besides, C4.5 algorithm gives most effective performance with large amount of datasets. Hence, the advantages of C4.5 algorithm are considerable but a dramatic increase in demand for large data would be improved to meet its performance.

The C4.5 algorithm builds a decision tree by learning from a training set in which every sample is built on an attribute-value pair. The current attribute node is calculated based on the information gain rate in which the root node is selected based on the extreme

| Algorithm 1 Pseudo code for C4.5 decision tree algorithm. |
| --- |

**Input:** Dataset
**Output:** Decision tree
// **Start**
      **for all** attributes in data
            Calculate information gain
      **end**
            **HG**= Attribute with highest information gain
            **Tree** = Create a decision node for splitting attribute **HG**
            **New data**= Sub datasets based on HG
      **for all** New data
            **Tree new**= C4.5(New data)
            Attach tree to corresponding branch of Tree
      **end**
      **return**

information gain rate. The data is numeric with only the classification as nominal leading category of labeled dataset. Hence, it is necessary to perform supervised data mining on the targeted dataset. This reduces the choice of classifiers in which a pre-defined classification could handle numerical data and classification in decision tree application. Each attribute is evaluated to find its ratio and rank during the learning phase of decision trees. Additionally, correlation coefficient is found to investigate the correlation between attributes as some dataset could not give any relevant result in data mining. In C4.5 decision tree algorithm, the gain is optimized by proposed SHO technique. The information gain is a rank based approach to compute the entropy. In this algorithm, the node with a highest normalized gain value is allowed to make decision, so there is a need to tune the gain parameter. The gain fitness is calculated based on the difference between actual gain value and new gain value. This is the objective function of the gain optimization technique which is described in Eq. (8).

$$fitness = \min\left\{ G_i - \hat{G}_i \right\} \tag{8}$$

Here, $G_i$ and $\hat{G}_i$ denotes actual and new gain, respectively. Based on this fitness, the gain error is minimized by SHO and the gain value will be computed by using Eq. (5). SHO can improve the learning accuracy, remove the redundant features and update the weight function of decision trees. The feature of SHO is random initialization generating strategy.

## PROPOSED METHOD: SHO

Selfish Herd Optimization is utilized to minimize the gain error in a better way in optimization process. It improves the balancing between exploration and exploitation phase without changing the population size (*Fausto et al., 2017*). SHO algorithm is mainly suitable for gain optimization in decision trees. In meta-heuristic algorithms, SHO is a new branch inspired from group dynamics for gain optimization. SHO is instigated from the simulations of herd and predators searching their food or prey. The algorithm uses search agents moving in n-dimensional space to find solution for optimization problem. The populations of SHO are herd and predators where the individuals are known as search

agents. In optimization areas, SHO is proved to be competitive with particle swarm optimization (PSO) (*Fausto et al., 2017*) for many tasks. The theory of Selfish Herd has been establishing the predation phase. Every herd hunts a possible prey to enhance the survival chance by accumulating with other conspecifics in ways that could increase their chances of surviving a predator attack without regard for how such behavior affects other individuals' chances of survival. This may increase the likelihood of a predator escaping from attacks regardless of how such activities disturb the survival probabilities of other individuals. The proposed SHO algorithm consists of different kinds of search agents like a flock of prey that lives in aggregation (mean of selfish herd), package of predators and predators within the said aggregate. This type of search agents is directed separately through fixed evolutionary operators which are centered on the relationship of the prey and the predator (*Anand & Arora, 2020*). The mathematical model of SHO algorithm is given as follows.

### Initialization

The iterative process of SHO's first step is to initialize the random populations of animals as prey and predators thereby having one set of separable locations $S = \{s1, s2, ...sN\}$. Here, the population size is denoted by $N$. The position of animals is limited into lower and upper boundaries and the groups are classified into two, like prey and predator. Eq. (9) is utilized to calculate the number of members in prey group.

$$n_p = \text{floor}(n \times \text{rand}(0.7, 0.9)) \tag{9}$$

Here, the quantity of prey group members is denoted as $n_p$ where $n$ denotes the population of the prey and the predators. In SHO, the number of prey (herd's size) is randomly selected within range 70% and 90% of the total population $n$, while the remainder individuals are labeled as predators. Therefore, 0.7 and 0.9 were the selected random values.

### Assignation of survival value

The survival value (SV)of every animal is assigned and it is associated with the current best and worst positions of a known $SV$ of whole population members. By optimization process, the present best and worst values are mentioned in the optimization problem. Then, the SV will be determined by using Eq. (10).

$$SV = \frac{f(x_i) - f_b}{f_w - f_b} \tag{10}$$

where, worst and best fitness values are denoted by $f_w$ and $f_b$, respectively. Here, $x_i$ represents the location of the prey or the predator.

### Herd's leader movement

All herd members' movement is one of the significant steps in SHO. The location of leader of the herd is updated by Eq. (11) as given in *Fausto et al. (2017)*.

$$h_L = \begin{cases} h_L + 2 \times r \times \phi_{l,P_m} \times (P_m - h_m) & if \; SVh_L = 1 \\ h_L + 2 \times r \times \psi_{l,ybest} \times (y_{best} - h_L) & if \; SVh_L < 1 \end{cases} \quad (11)$$

Here, the tested selfish repulsion towards predators by current herd leader is denoted as $\phi_l$, and r denotes the random number in the range (0, 1). $h_L$, $h_m$ and $p_m$ are indicated as herd leader, herds center of mass and predators center of mass, respectively. $\psi_L$ Indicates the selfish attraction examined by the leader of the flock toward the global best location $y_{best}$.

Moreover, the location of the herd member $h_a$ is updated based on two selections. Equation (12) is utilized to follow the herd and Eq. (14) is utilized to recompense the group. Also, the selection is prepared based on some random variables.

$$h_a = h_a + f_a \quad (12)$$

where,

$$f_a = \begin{cases} 2 \times \left( \beta \times \psi_{h_a,h_L} \times (h_L - h_a) + \gamma \times \psi_{h_a,h_b}(h_b - h_a) \right) & SV_{h_L} \leq SV_{h_u} \\ 2 \times \delta \times \psi_{h_i,h_m} \times (h_m - h_a) & otherwise \end{cases} \quad (13)$$

$$h_a = h_a + 2 \times \beta \times \psi_{h_L,y_{best}} \times (y_{best} - h_a) + \gamma \times (1 - SV_{h_a}) \times \hat{r} \quad (14)$$

Here, $\psi_{h_a,h_m}$ and $\psi_{h_a,h_L}$ indicates the selfish attractions examined through the herd member $h_a$ towards $h_b$ and $h_L$, while $\beta$, $\gamma$ and $\delta$ indicates the random numbers in the range (0, 1) and present herds' leader position is denoted as $h_b$. Also, $\hat{r}$ represents the random direction unit vector.

## Predator movement

The movement of every separable set of predators, the endurance of entities in the attacked flock and the distance between the predators from assault predators are taken into account in SHO. Based on the pursuit probability, the predator movement is determined as given in Eq. (15).

$$P_i = \frac{\varpi_{pi,j_j}}{\sum_{m=1}^{N_h} \varpi_{pi,j_j}} \quad (15)$$

The prey attractiveness amongst $p_i$ and $h_j$ is denoted as $\varpi_{pi,j_j}$. Then the predator position $X_P$ is updated by Eq. (16).

$$X_p = X_P + 2 \times r \times (h_r - X_p) \quad (16)$$

where, $h_r$ indicates randomly chosen herd member. In advance, each member of the predator and the prey group survival rate is recomputed by Eq. (9).

## Predation phase

The predation process is executed in this phase. Domain danger is defined by SHO which is signified as area of finite radius around each prey. The domain danger radius $R_r$ of each prey is computed by Eq. (17).

$$R_r = \frac{\sum_{j=1}^{n}}{\left| y_j^l - y_j^u \right|} \tag{17}$$

where, upper and lower boundary members are represented by $y_j^u$ and $y_j^l$, respectively and the dimensions are denoted as $n$. After the radius calculation, a pack of targeted prey is computed by Eq. (18).

$$T_{p_i} = h_j \in H | SV_{h_j} < SV_{p_i} \| P_i - h_j \| \leq R_r, h_j \notin K \tag{18}$$

Here, $SV_{h_j}$ and $SV_{p_i}$ denotes the endurance tenets of $P_i$ and $h_j$ correspondingly. $\| pi - h j \|$ signifies the Euclidean distance amongst the entities $P_i$ and $h_i$, respectively. Also the herds' population is denoted as $H$. The probabilities of the existence hunted are computed for every member of the set and is formulated in Eq. (19) where $K$ is set of killed herd members $\{ K = K, h_j \}$.

$$H_{p_i,h_j} = \frac{\varpi_{p_i,h_j}}{\sum_{(h_m \in T_{p_i})} \varpi_{p_i,h_m}}, \; h_j \in T_{p_i} \tag{19}$$

### Restoration phase

Finally, the restoration is accomplished by making a set $M = h_j \notin K$. Here, $K$ represents the set of herd member slayed for the duration of the predation phase. The mating probabilities are also determined by each member as in Eq. (20),

$$P_r = \frac{SV_{h_j}}{\sum_{(h_m \in M)} SV_{h_m}}, \; h_j \in M \tag{20}$$

Each $h_j \in K$ is changed by a different result by SHO's mating operation which is $mix([h_{r1}, h_{r2}, \ldots h_{rn}])$. This SHO algorithm is utilized to optimize the gain function in data classification operation. Figure 1 displays the flow diagram of SHO algorithm.

## RESULT AND DISCUSSION

The efficiency of our proposed method is assessed by comparing its accuracy with other popular classification methods like PSO (*Chen et al., 2014*), ACO (*Otero, Freitas & Johnson, 2012*) and Cuckoo Search (CS) Optimization (*Cao et al., 2015*). We estimated the performance of proposed algorithm based on the accuracy as tested in 10 UCI datasets. The accuracy of our proposed method is comparable to other optimization methods and various classifiers. But the cross validation is not performed in the proposed approach. The proposed method is greater than all existing methods taken for comparison. SHO is combined with C4.5 classifier to produce greater accuracy than a standard C4.5 classifier. The proposed decision tree classifier named C4.5-SHO is further compared with C4.5, ID3 and CART. The description of ten data sets is tabulated in Table 1. These datasets include Monks, Car, Chess, Breast-cancer, Hayes, Abalone, Wine, Ionosphere, Iris and Scale (*Arellano, Bory-Reyes & Hernandez-Simon, 2018*). Table 2 shows the algorithm parameters. Table 3 shows the algorithm parameters for decision tree.

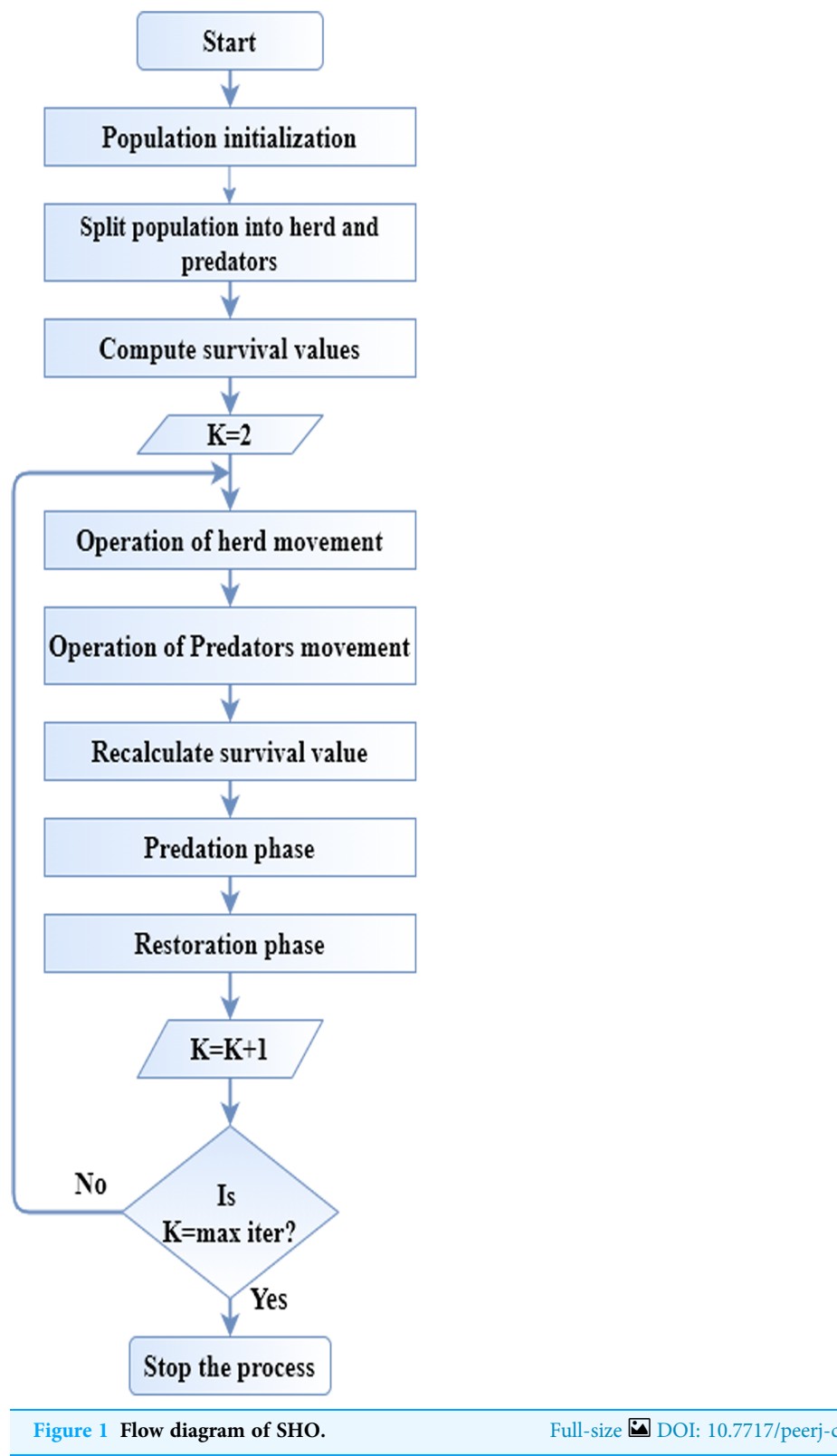

**Figure 1  Flow diagram of SHO.**   

---

**Algorithm 2** Peseudo code for the proposed SHO algorithm in data classification.

**Start**
Initialize the parametrs and locations of SHO by Eq. (9)
    **For**
        Each individual
        Compute survival by Eq. (10)
    **End for**
    **While** $K < K_{max}$
        **For** every prey movement
          **If** prey's leader
            Update the location of prey leader by Eq. (11)
          **Else**
            Update prey location by Eq. (14)
          **End if**
        **End for**
        **For**
          Every predator's movement
          **For** each prey
            Determine predation probabitity Eq. (15)
          **End for**
          Update predator location by Eq. (16)
        **End for**
          Re-compute survival value using Eq. (10)
          Compute dangerous radius by Eq. (17)
          Predation performance by Eqs. (18) and (19)
          Restoration performance by Eq. (20)
        $K = K + 1$
    **End while**
    Global optimal output
    Fitness for global optimal output
**End**

---

The proposed method is compared with existing entropies, optimization algorithms and different classifiers. The effectiveness is estimated based on the accuracy, AUROC and classifier.

**(a) Accuracy**

The classification accuracy is measured based on Eq. (21) (*Polat & Güneş, 2009*),

$$accuracy\,(A) = \frac{\sum_{i=1}^{|A|} assess(a_i)}{|A|},\; a_i \in A \tag{21}$$

$$assess(a) = \begin{cases} 1, & \text{if } classify(a) = a.c \\ 0, & \text{otherwise} \end{cases} \tag{22}$$

Here, $A$ is denoted as the dataset to be classified (test set) $a \in A$, $a.c$ is the class of item $a$ and classify $(a)$ returns the classification through C4.5 classifier.

In Table 4, the proposed C4.5-SHO decision tree classification accuracy is compared with other classifiers like C4.5, ID3 and CART. The accuracy of our proposed work is more stable compared to the accuracy achieved by the other considered algorithms.

**Table 1 Description of data set.**

| Data set | No. of attributes | No. of samples | Classes |
|---|---|---|---|
| Monks | 7 | 432 | 2 |
| Car | 6 | 1,728 | 4 |
| Chess | 6 | 28,056 | 36 |
| Breast-cancer | 10 | 699 | 2 |
| Hayes | 5 | 160 | 3 |
| Abalone | 8 | 4,177 | 2 |
| Wine | 13 | 178 | 2 |
| Ionosphere | 34 | 351 | 2 |
| Iris | 4 | 150 | 2 |
| Scale | 4 | 625 | 2 |

**Table 2 Algorithms parameters and values.**

| SHO | | ACO | | PSO | | CS | |
|---|---|---|---|---|---|---|---|
| Number of populations | 50 | Number of populations | 50 | Number of populations | 100 | Number of populations | 50 |
| Maximum iterations | 500 | Maximum iterations | 500 | Maximum iterations | 500 | Maximum iterations | 500 |
| Dimension | 5 | Phromone exponential weight | −1 | Inertia weight | −1 | Dimension | 5 |
| Lower boundary | −1 | Heuristic exponential weight | 1 | Inertia weight damping ratio | 0.99 | Lower bound and upper bound | −1 and 1 |
| Upper boundary | 1 | Evaporation rate | 1 | Personal and global learning coefficient | 1.5 and 2 | Number of nests | 20 |
| Prey's rate | 0.7, 0.9 | Lower bound and upper bound | −1 and 1 | Lower bound and upper bound | −10 and 10 | Transition probability coefficient | 0.1 |
| Number of runs | 100 | Number of runs | 100 | Number of runs | 100 | Number of runs | 100 |

The accuracy of classification is depended on the training dataset. The dataset is split up into a training set and test set. The classifier model is trained with training set. Then to evaluate the accuracy of the classifier, we use test set to predict the labels (which we know) in the test set. The accuracy of Iris data set is high (0.9986) compared to other data sets. The lowest accuracy of the proposed C4.5-SHO is 0.9437 in Scale data set. In comparison with existing classifiers, it is observed that the proposed approach has obtained a good accuracy.

In Table 5, the proposed C4.5-SHO decision tree classification accuracy is compared with other algorithms like ACO, PSO and CS. The accuracy of our proposed work is more stable compared to the accuracy achieved by the other considered algorithms. The accuracy of Iris data set is high (0.9986) compared to other data sets. The lowest accuracy of the proposed C4.5-SHO is 0.9437 in Scale data set. In comparison with existing algorithms, the proposed approach achieved good accuracy.

**Table 3 Algorithms parameters for decision tree.**

| C4.5 | | ID3 | | CART | |
|---|---|---|---|---|---|
| Confidence factor | 0.25 | Minimum number of instances in split | 10 | Complexity parameter | 0.01 |
| Minimum instance per leaf | 2 | Minimum number of instances in a leaf | 5 | Minimum number of instances in split | 20 |
| Minimum number of instances in a leaf | 5 | Maximum depth | 20 | Minimum number of instances in a leaf | 7 |
| Use binary splits only | False | – | | Maximum depth | 30 |

**Table 4 Classification accuracy of the proposed classifier C4.5 with C4.5, ID3 and CART.**

| Data set | C4.5-SHO | C4.5 | ID3 | CART |
|---|---|---|---|---|
| Monks | 0.9832 | 0.966 | 0.951 | 0.954 |
| Car | 0.9725 | 0.923 | 0.9547 | 0.8415 |
| Chess | 0.9959 | 0.9944 | 0.9715 | 0.8954 |
| Breast-cancer | 0.9796 | 0.95 | 0.9621 | 0.9531 |
| Hayes | 0.9553 | 0.8094 | 0.9014 | 0.7452 |
| Abalone | 0.9667 | 0.9235 | 0.9111 | 0.9111 |
| Wine | 0.9769 | 0.963 | 0.9443 | 0.9145 |
| Ionosphere | 0.9899 | 0.9421 | 0.9364 | 0.9087 |
| Iris | 0.9986 | 0.9712 | 0.7543 | 0.8924 |
| Scale | 0.9437 | 0.7782 | 0.7932 | 0.7725 |
| Average value | 0.97623 | 0.92208 | 0.908 | 0.87884 |

**Table 5 Classification accuracy of the proposed Algorithm with ACO, PSO and CS.**

| Data set | SHO-C4.5 | ACO | PSO | CS |
|---|---|---|---|---|
| Monks | 0.9832 | 0.9600 | 0.9435 | 0.9563 |
| Car | 0.9725 | 0.9322 | 0.9298 | 0.9202 |
| Chess | 0.9959 | 0.9944 | 0.9944 | 0.9742 |
| Breast-cancer | 0.9796 | 0.9555 | 0.954 | 0.9621 |
| Hayes | 0.9553 | 0.90311 | 0.9322 | 0.9415 |
| Abalone | 0.9667 | 0.9500 | 0.9345 | 0.9247 |
| Wine | 0.9769 | 0.9240 | 0.8999 | 0.8924 |
| Ionosphere | 0.9899 | 0.9583 | 0.9645 | 0.9645 |
| Iris | 0.9986 | 0.9796 | 0.9741 | 0.9764 |
| Scale | 0.9437 | 0.9060 | 0.9177 | 0.8911 |
| Average value | 0.97623 | 0.946311 | 0.94446 | 0.94034 |

**(b) Area under ROC**

The performance of classification model is shown through graph analysis of AUROC. This is dependent upon the attributes as well as classes. The proposed C4.5-SHO is compared with other classifiers like C4.5, ID3 and CART. The AUROC results presented in Table 6 which shows that the AUROC value of proposed method is better than other algorithms.

**Table 6 Area under the ROC curve of proposed C4.5 with ID3 and CART.**

| Dataset | C4.5-SHO | C4.5 | ID3 | CART |
|---|---|---|---|---|
| Monks | 0.9619 | 0.95713 | 0.9636 | 0.9791 |
| Car | 0.9819 | 0.9393 | 0.9891 | 0.8933 |
| Chess | 0.9673 | 0.9252 | 0.9090 | 0.9049 |
| Breast-cancer | 0.9793 | 0.9171 | 0.9730 | 0.9218 |
| Hayes | 0.9874 | 0.9069 | 0.9108 | 0.8360 |
| Abalone | 0.9647 | 0.9224 | 0.9573 | 0.9082 |
| Wine | 0.9914 | 0.9772 | 0.9497 | 0.9739 |
| Ionosphere | 0.9943 | 0.9680 | 0.9059 | 0.9560 |
| Iris | 0.9890 | 0.9048 | 0.7945 | 0.9481 |
| Scale | 0.9850 | 0.8562 | 0.7845 | 0.8007 |
| Average value | 0.98022 | 0.92742 | 0.91374 | 0.9122 |

**Table 7 Area under ROC curve of the proposed Algorithm with ALO, PSO and CS.**

| Dataset | C4.5-SHO | ACO | PSO | CS |
|---|---|---|---|---|
| Monks | 0.9935 | 0.9874 | 0.97668 | 0.9733 |
| Car | 0.98452 | 0.97908 | 0.97583 | 0.9659 |
| Chess | 0.99931 | 0.98612 | 0.9815 | 0.9503 |
| Breast-cancer | 0.9854 | 0.9795 | 0.9695 | 0.9581 |
| Hayes | 0.99616 | 0.92611 | 0.9442 | 0.9571 |
| Abalone | 0.9885 | 0.9828 | 0.9694 | 0.9566 |
| Wine | 0.9932 | 0.9830 | 0.8977 | 0.8964 |
| Ionosphere | 0.9954 | 0.9741 | 0.9630 | 0.9569 |
| Iris | 0.9873 | 0.9687 | 0.9656 | 0.9578 |
| Scale | 0.9858 | 0.9266 | 0.9165 | 0.8968 |
| Average value | 0.9909 | 0.96934 | 0.95599 | 0.94692 |

The proposed C4.5-SHO is compared with other optimization algorithms like ACO, PSO and CS. The AUROC results are presented in Table 7 which shows that the proposed AUROC value is better than existing algorithms. It is revealed that SHO not only reduces the complexity of decision trees but also enhances the accuracy.

**(c) Different entropy comparison**

Based on the Ray's quadratic entropy, the information gain is optimized through SHO algorithm. The entropy with SHO is compared to traditional SHO in terms of other entropies, such as C4.5-SHO (Shanon entropy), C4.5–SHO (Havrda and charvt entropy), C4.5- SHO (Renyi entropy) and C4.5- SHO (Taneja entropy). The quadratic entropy is the measure of disorder in the range between entire arranged (ordered) and unarranged (disordered) data in the given dataset. The quadratic entropy is successfully measured for the disorders in the datasets. The classification accuracy is improved by the quadratic

**Table 8 Entropy comparison.**

| Dataset | C4.5-SHO (Shanon entropy) | C4.5 – SHO (Havrda & charvt entropy) | C4.5 – SHO (Quadratic entropy) | C4.5- SHO (Renyi entropy) | C4.5- SHO (Taneja entropy) |
|---|---|---|---|---|---|
| Monks | 0.9429 | 0.9756 | 0.9859 | 0.9926 | 0.9415 |
| Car | 0.9585 | 0.9527 | 0.9753 | 0.9895 | 0.9700 |
| Chess | 0.9510 | 0.9535 | 0.9907 | 0.9809 | 0.9401 |
| Breast-cancer | 0.9852 | 0.9558 | 0.9863 | 0.9564 | 0.9672 |
| Hayes | 0.9579 | 0.9460 | 0.9981 | 0.9476 | 0.9102 |
| Abalone | 0.9556 | 0.9618 | 0.9789 | 0.9715 | 0.9447 |
| Wine | 0.9485 | 0.9731 | 0.9823 | 0.9297 | 0.9317 |
| Ionosphere | 0.9319 | 0.9415 | 0.9665 | 0.9636 | 0.9036 |
| Iris | 0.9465 | 0.9807 | 0.9832 | 0.9514 | 0.9428 |
| Scale | 0.9725 | 0.8936 | 0.9747 | 0.9617 | 0.9031 |
| Average Value | 0.95505 | 0.95343 | 0.98219 | 0.96449 | 0.93549 |

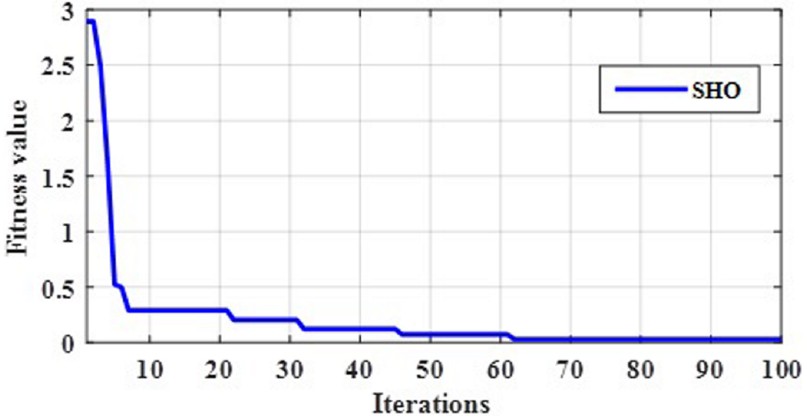

**Figure 2 Convergence evaluation of SHO.**

entropy than other entropies. Hence, the proposed work follows Ray's quadratic entropy to get a better output. Compared to other entropies, the quadratic entropy achieved better accuracy in data classification for all data sets. Table 8 shows the entropy comparisons with proposed SHO.

The gain parameter is optimized by proposed C4.5-SHO algorithm in order to make a decision. An optimal gain value is selected through the fitness function mentioned in Eq. (8). Initially, gain is calculated for each attribute used in the decision tree. If the number of iteration increases, the gain value will be changed on every iteration. Further, the fitness is nothing but the difference between actual gain and new gain. Therefore, the gain values of the attributes are noted for every iteration. The proposed optimization algorithm provided the optimal best gain value at 100th iteration as seen in the convergence plot in Fig. 2. Finally, the gain error was minimized with the help of C4.5-SHO algorithm.

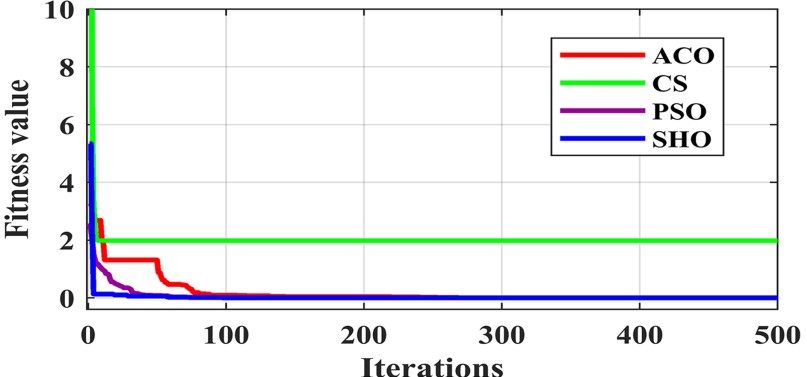

**Figure 3** Comparison of convergence plot.

**Table 9 Computational time.**

| Algorithm | Time (sec) |
| --- | --- |
| ACO | 0.974 |
| PSO | 0.54 |
| CS | 0.6 |
| SHO | 0.49 |

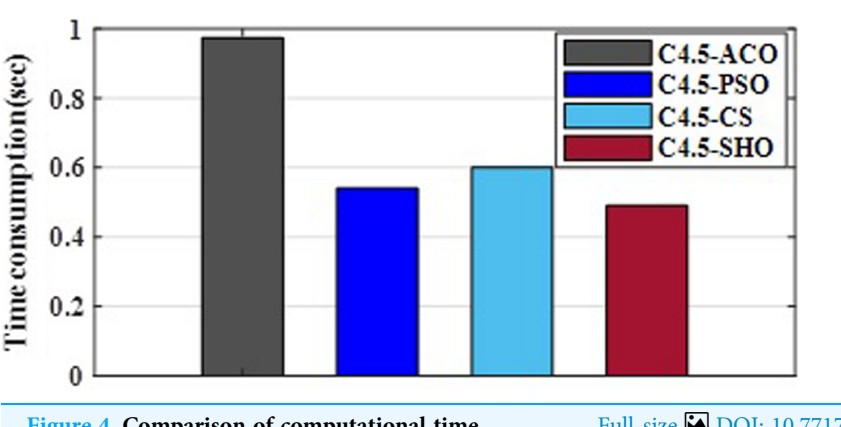

**Figure 4** Comparison of computational time.

Figure 3 illustrates the convergence plot of proposed SHO and similar existing algorithms for average of all datasets. The proposed SHO achieved good convergence compared to existing techniques. The proposed work is based on gain optimization with SHO algorithm whereas the execution time is also the most important factor in data classification approach. On comparing the time-taken for analysis, the proposed method needs low computational time than the existing algorithms like ACO (0.974s), PSO (0.54s) and CS (0.6s). Table 9 and Fig. 4 illustrate the computational time comparison for average of all datasets.

## CONCLUSION

Data mining is a broad area that integrates techniques from several fields including machine learning, statistics, artificial intelligence, and database systems for the analysis of a large amount of data. This paper presented a gain optimization technique termed as C4.5-SHO. The effectiveness of quadratic entropy is estimated and discussed to evaluate the attributes in different datasets. This article presents the most influential algorithms for classification. The gain of data classification information is optimized by the proposed SHO algorithm. The evaluation of C4.5 decision tree based SHO results show that the AUROC is the best measure because of the classification of unbalanced data. The accuracy of proposed C4.5-SHO technique is higher than the existing techniques like C4.5, ID3 and CART. The proposed approach is compared with the algorithms of ACO, PSO and CS for AUROC. A better accuracy (average 0.9762), better AUROC (average 0.9909) and a better computational time (0.49s) are obtained from the gain optimized technique of C.5-SHO. In future, hybrid optimization technique is utilized to improve the data classification information gain.

### Funding
The authors received no funding for this work.

### Competing Interests
The authors declare that they have no competing interests.

### Author Contributions
- G. Sekhar Reddy conceived and designed the experiments, performed the experiments, analyzed the data, performed the computation work, prepared figures and/or tables, authored or reviewed drafts of the paper, and approved the final draft.
- Suneetha Chittineni conceived and designed the experiments, performed the experiments, analyzed the data, performed the computation work, prepared figures and/or tables, authored or reviewed drafts of the paper, and approved the final draft.

### Data Availability
Raw measurements and MATLAB codes are available in the Supplemental Files.

### Supplemental Information
Supplemental information for this article can be found online at http://dx.doi.org/10.7717/peerj-cs.424#supplemental-information.

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
