# Peer review of "Entropy based C4.5-SHO algorithm with information gain optimization in data mining"

_PeerJ Computer Science, doi:10.7717/peerj-cs.424_

## Round 0.1 · original submission · Major Revisions

As per the feedback by the reviewers, the authors need to address several major comments regarding the experimental design and the level of provided details. I suggest you address the comments and resubmit in a timely manner.

Reviewer 1 ·

Basic reporting

no comment

Experimental design

no comment

Validity of the findings

no comment

Additional comments

This paper describes a new classification algorithm to improve information management. The use of the SHO algorithm has a significant role to produce optimal values. This paper also explains that the proposed method has a better level of accuracy than the previous methods.
This paper is well written. Gap research is well presented. The proposed method can be described systematically. Unfortunately, there are only a few small errors but it is quite annoying, i.e., in Figure 1. The SHO algorithm is not well illustrated in the end. If this can be corrected, it will make this paper of higher quality.

·

Basic reporting

The manuscript is overall well written and documented. However, I have the following comments:
Lines 53-56. I suggest to re-write these sentences as a reader may understand that sensitivity is the only important metric. Or that the models can easily predict the minority classes.

Lines 63-64. I suggest to modify the sentence as parameters are still needed for decision trees, e.g., purity, minimum number of samples in a node, etc.

Line 78-79. I suggest to move the sentence introducing decision tree to previous paragraph, lines 57-59.

Line 166. Define the acronyms before using them. What is RLBOR?

Paragraph 174. Sentence: “C4.5 is one of the best…”. Add references and justification to prove this affirmation.

Paragraph 179. Sentence: “The decision tree trained process is enormous and deep.” Please explain why, and compared to what?

Lines 184-187. Please clarify that these disadvantages (e.g., large data) may only happen on some applications. Decision trees are among the fastest models to train compared to other models (ANN, SVM, etc.).

Algorithm 1. First loop, “for i=1: number of data”, number of data samples? If so, should this not loop through the features? That is, calculate the information gain for each feature.

Algorithm 1. Second for loop is also confusing, should this not be a recursive process? That is, function calling itself twice with the left and right partition of the data.

Paragraph 188. I am not sure I understand the sentence “The current attribute node is one and…”.

Paragraph 188. Sentence: “It is necessary to perform supervised data mining on the targeted dataset”. Not sure I understand why is it necessary? For which applications? For deriving decision trees?

Line 194. Please indicate why is SHO better and why, compared to what? Add references.

Line 199. PSO abbreviation not defined. Also, add references to prove the remark.

Lines 243, 257, and 266. Fix the number of subsections. They all have 1.

Line 267. Can you please explain why Ray’s quadratic entropy is better? Compared to
what and why? Maybe this sentence can be stated after explaining the results from Table 6.

Figures 2 and 3. Please increase the resolution. Text is not readable.

The following are a few grammar suggestions.
1. Use of comma before respectively, e.g., “x, y, and z, respectively.”
2. Capitalize each letter after “.” punctuation mark. Example, line 133.
3. Avoid starting sentences with the “It” pronoun. Although the contexts for most sentences are clear, others may be confusing for a reader. Lines 115, 124, 131, 140, 144, etc. For example, the sentence in paragraph 179: “It measures the information related to classification obtained on the basis of the same prediction”. Does this sentence refer to gain ratio or information gain? Also, in that same sentence, change “form” to “from”.
4. Add a comma before “such as”. Lines 21, 32, etc.
5. Use a comma before which when it introduces a nonrestrictive phrase. Don't use a comma before which when it's part of a prepositional phrase, such as “in which.”
6. Please use ID3 abbreviation consistently. The manuscript has Id3 in different paragraphs.
7. Data is plural, please replace “data is” by “data are” in the manuscript.
Finally, please ensure references follow the PeerJ reference style.

Experimental design

The experimental design is well performed. However, I have the next suggestions:

Equation 9. Why rand(0.7,0.9)?
Equation 11. Define all variables used, e.g., hl and hm.
Equation 12. Define hb.
Equation 16. Define hr.
Equation 17. Define Rr, i.e., the radius is calculated for each pray or pack of prays?
Equation 18. Define K and H.

I would suggest adding an additional paragraph and possibly a figure explaining how the SHO is implemented for the optimization of the gain. Moreover, fitness defined in equation 8 considers only the gain at a specific node. How is SHO used to optimize all gain for splitting non-terminal nodes? Is it required to run one SHO per non-terminal node?

Line 235. Although the manuscript mentions that cross-validation was used, there are no details about the tuning of the parameters for the different models. Please define if default parameters were used or if these are tuned. If tuned, which technique did you consider to tune them (grid search, random search, etc.) and which data used (did you use validation set?). Finally, mention that 10-fold cross-validation was used.

Can you please describe the parameters used for SHO, ACO, lower/upper bounds, etc.? That is, how many iterations, trees, etc. were used, and why? Where these set randomly or optimized?

Validity of the findings

Results are well described. I would only suggest adding an average row in Tables 2-6 to better observe the overall performance improvement. Additionally, highlighting in bold the best performance obtained for each dataset would increase readability.

Fig. 2 and Table 7. Are convergence and computational time shown for a single dataset or an average?

Can you please explain why RUN.m uses k=100 for the number of trees? Are you reporting the best results obtained from these 100 SHO runs?

While the code is very well documented, it only includes the SHO implementation. For reproducibility purposes, I would suggest providing the other optimization algorithms (ACO and PSO) if implemented. Otherwise, refer to the code/source used.

---

## Round 0.2 · Minor Revisions

After careful consideration of the reviewer's comments, I recommend authors address the points and respond in a timely manner. In particular, more details regarding experiments need to be provided. Acceptance of the work will be contingent on sufficiently addressing the points raised in this second revision.

·

Basic reporting

Caption Table 4. Replace “ALO” by “ACO”
Few grammar suggestions:

Line 228. Replace “So, chose 0.7 and 0.9 values as the random values.” By “Therefore, 0.7 and 0.9 were the selected random values.”

Line 236. Use comma before respectively.

Line 274, and 268. Replace “The accuracy of our proposed work is almost stable than the other.” By “The accuracy of our proposed work is more stable compared to the accuracy achieved by the other considered algorithms.”

Line 269. Can you please explain what do you mean by: “The accuracy of classification is depended on the training dataset.”. This phrase may be interpreted as the reported results are those obtained from the training data.

Experimental design

Refer to the validity of findings section.

Validity of the findings

Below I repeat my original comment, Authors reply, and my new comment:

Original reviewer’s comment: Can you please explain why RUN.m uses k=100 for the number of trees? Are you reporting the best results obtained from these 100 SHO runs?

Author’s reply: k=100 is for the number of times the tree is generated. RUN.m file is run 100 times. After 100 SHO runs, we have attain the best result that result is presented in the article.

Second revision – reviewer’s comment: Ideally, data would need to be split into training, validation, and test sets. A validation set is used to tune the parameters. In this case, if you ran 100 executions, the best performing model (based on validation set) is then used to test the performance of the final model by using the test set. What I am trying to say here, the execution of those 100 different trees is considered to be part of the training phase.
Some studies reserve the validation from the training set, while other use nested cross-validation (see https://stats.stackexchange.com/questions/103828/use-of-nested-cross-validation).
My concern is that other algorithms (ACO, PSO, and CS) were not given the same opportunities to achieve better results if these algorithms were not run 100 times too. Arguably, you would be able to continue to improve results on test data just by expanding the number of executions (that is why these iterations should be considered as the training phase). These models would overfit the test data.

With respect to the other Author’s response: “The parameters of SHO, ACO, PSO, and CS values are tabulated in Table 2. The parameter table is updated in the revised manuscript. We have taken random values for our proposed optimization work like iterations and number of populations”

These number of iterations and populations should be part of the training phase which is optimized by using the validation set. Final results would be those obtained by testing the final model on the test data that has not been seen during any training phase.
If validation set was not used to compare C4.5-SHO against C4.5, ID3, and CART. Then authors need to carefully indicate the parameters used for each of these algorithms to demonstrate that all were given the same opportunities and indicate so in the manuscript. The authors should indicate that a validation set was not used.
Similarly, I suggest authors describe the search space for the other C4.5 implementations (C4.5 ACO, C4.5 PSO, and C4.5 CS), i.e., where they also ran 100 times? Was a random search also performed?

Additional comments

I thank the authors for the modifications.
While the Authors answered most of my comments. I still have a concern related to the optimization of parameters (optimization algorithms and decision tree algorithms).

---

## Round 0.3 · accepted · Accept

In this second review, the authors have addressed sufficiently the previous minor comments of the reviewer. The authors have included the required details in the final revised version.